# SeaKR: Self-aware Knowledge Retrieval for Adaptive Retrieval Augmented Generation

## Abstract

Adaptive Retrieval-Augmented Generation (RAG) is an effective strategy to alleviate hallucination of large language models (LLMs). It dynamically determines whether LLMs need external knowledge for generation and invokes retrieval accordingly. This paper introduces *Self-aware Knowledge Retrieval* (SeaKR), a novel adaptive RAG model that extracts self-aware uncertainty of LLMs from their internal states. SeaKR activates retrieval when the LLMs present high self-aware uncertainty for generation. To effectively integrate retrieved knowledge snippets, SeaKR re-ranks them based on LLM's self-aware uncertainty to preserve the snippet that reduces their uncertainty to the utmost. To facilitate solving complex tasks that require multiple retrievals, SeaKR utilizes their self-aware uncertainty to choose among different reasoning strategies. Our experiments on both complex and simple Question Answering datasets show that SeaKR outperforms existing adaptive RAG methods.

## 1 Introduction

Retrieval-Augmented Generation (RAG, Lewis et al., 2020; Gao et al., 2023) retrieves and integrates external knowledge into the context of large language models (LLMs, Achiam et al., 2023; Touvron et al., 2023; Meta, 2024). RAG represents a promising strategy to combat the issue of hallucination Trivedi et al. (2022); Yao et al. (2022); Ji et al. (2023); Cao et al. (2023)—where LLMs produce factually incorrect answers camouflaged as correct ones—primarily caused by queries that exceed the limited parametric knowledge boundaries Yin et al. (2024) of LLMs.

Most existing RAG methods retrieve knowledge for every input query by default. However, due to the noisy nature of the data storage, retrieved knowledge can be misleading or even conflicting when the LLM can extract the correct answer from its own parametric knowledge Mallen et al. (2022); Xie et al. (2023); Liu et al. (2024). Conducting retrieval for every generation is both inefficient and unnecessary.

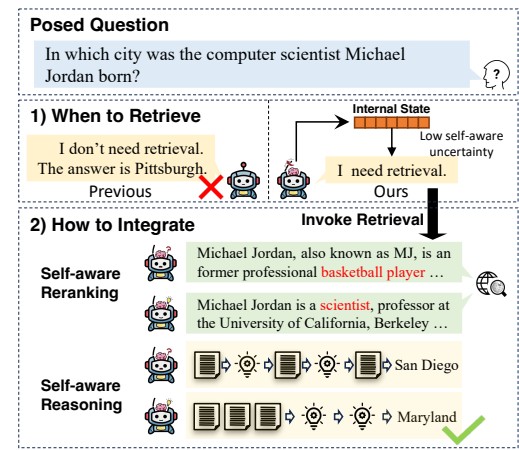

Figure 1: Adaptive RAG mainly concerns 1) when to retrieve and 2) how to integrate retrieved knowledge.

Adaptive retrieval strategy Jiang et al. (2023); Su et al. (2024); Wang et al. (2023; 2024) is hence proposed to dynamically determine whether LLMs require external knowledge and then invoke the retrieval step accordingly.

Adaptive RAG needs to consider two major factors: *1) When to retrieve knowledge and 2) How to integrate retrieved knowledge.* Recent studies Kadavath et al. (2022); Zhu et al. (2023) show that LLMs are aware of their uncertainty for the generated content and this uncertainty can be discerned from their internal states Chen et al. (2023a); Zhang et al. (2024). We argue that this *self-aware*

nature of LLMs can be utilized to determine when retrieval is needed and help with knowledge integration. Motivated by this, we propose *SElf-Aware Knowledge Retrieval* (SEAKR) for adaptive RAG. To the best of our knowledge, SEAKR is the first to leverage self-awareness from the internal states of LLMs to dynamically determine when to retrieve and effectively integrate retrieved knowledge.

To decide *when to retrieve*, existing adaptive RAG Wang et al. (2024); Jiang et al. (2023); Su et al. (2024) judges the knowledge sufficiency of LLMs solely based on their outputs, which is prone to ubiquitous self-bias of LLMs Xu et al. (2024). In contrast, SEAKR initiates retrieval self-aware uncertainty from the internal states of LLMs, which more accurately determines the knowledge demand. To be specific, the self-aware uncertainty of LLMs is extracted from the internal states in the feed-forward network (FFN) of each layer corresponding to the last generated token. The consistency measure across multiple generations to the same prompt is computed as the self-aware uncertainty score of LLMs, subsequently used for the retrieval decision and knowledge integration.

To *effectively integrate retrieved knowledge* into the generation process, which is largely neglected by previous adaptive RAG methods, SEAKR designs two adaptive integration strategies based on the LLM self-awareness: 1) *Self-aware re-ranking*. SEAKR asks the LLM to read multiple recalled snippets and selects the knowledge that reduces most of its uncertainty as the augmented context. 2) *Self-aware reasoning*. SEAKR supports iterative knowledge retrieval to gather multiple knowledge for answering complex questions. With multiple retrieved knowledge, SEAKR integrates different reasoning strategies, including direct generation and comprehensive reasoning, to digest the knowledge. It then selects the strategy that produces the least generation uncertainty.

We conduct experiments on complex question-answering (QA) and simple QA tasks. We find that SEAKR brings substantial improvement over existing adaptive RAG methods on complex QA benchmarks. Our ablation study shows that dynamically integrating retrieved knowledge brings even more performance gain than self-aware retrieval, further highlighting the necessity of dynamical integration for adaptive RAG.

## 2 RELATED WORK

We formally define and introduce works related to SEAKR, including retrieval augmented generation and analyzing LLMs through their internal states.

### 2.1 RETRIEVAL AUGMENTED GENERATION

**Retrieval augmented generation** (RAG) system typically comprises a search engine for knowledge retrieval and a Large Language Model (LLM) for answer generation Khandelwal et al. (2019); Guu et al. (2020); Lewis et al. (2020); Borgeaud et al. (2022); Ram et al. (2023); Shi et al. (2023). Given a user-posed question, RAG first searches for relevant knowledge snippets using the search engine and then generates the answer via machine reading comprehension Chen et al. (2017).

**Adaptive retrieval augmented generation** dynamically determines whether LLMs require retrieved knowledge, thereby reducing the adverse effect of inaccurately retrieved information. FLARE Jiang et al. (2023) and DRAGIN Su et al. (2024) activate the search engine when LLMs output tokens with low probability. Self-RAG Asai et al. (2023) and Wang et al. (2024) prompt LLMs to decide on retrieval. Self-knowledge guided generation Wang et al. (2023) trains a classification model to judge the factuality of model generation.

Existing adaptive RAG methods mainly face two challenges. 1) To decide when to retrieve, it is superficial to have the decision of retrieval solely on the output of LLM. However, the retrieval decision made by LLMs is still at risk of hallucination, which potentially does not reliably indicate the actual knowledge sufficiency Yona et al. (2024). Furthermore, LLMs have the tendency to confidently produce incorrect contents even when correct knowledge is missing from their parameters Huang et al. (2023); Xu et al. (2024). 2) To integrate retrieved knowledge, these attempts rely on the correctness of search engine returned knowledge, neglecting to re-rank multiple retrieved knowledge and optimize the reasoning paths.

**Retrieval augmented reasoning** integrates the reasoning capabilities of LLMs into the RAG framework to solve complex questions. IRCoT Trivedi et al. (2022) implements retrieval augmentation

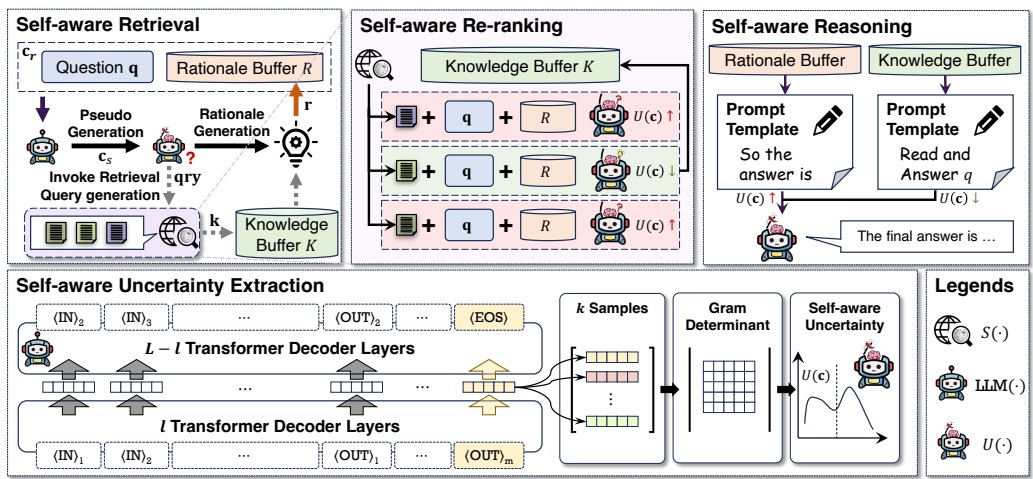

Figure 2: The overall framework of SEAKR.

within multi-step chain-of-thought (CoT, Wei et al., 2022) reasoning processes, which is adopted by many following works Su et al. (2024); Jeong et al. (2024). ProbTree Cao et al. (2023) decomposes complex questions into sub-questions, which are solved using RAG before being aggregated into the final answer.

## 2.2 Self-awareness in Internal States of LLMs

Most of the mainstream LLMs are stacks of Transformer Vaswani et al. (2017) decoders. To predict the next token, without losing generality, the $i^{\text{th}}$ layer processes the hidden representation $\mathbf{H}^{(l-1)}$ from its previous layer according to the formula: $\mathbf{H}^{(l)} = \texttt{FFN}\left(\texttt{Attn}\left(\mathbf{H}^{(l-1)}\right)\right)$, where $\texttt{Attn}(\cdot)$ denotes attention sub-layer, $\texttt{FFN}(\cdot)$ is the feed-forward sub-layer.

Many works Meng et al. (2022); Li et al. (2022); Gurnee & Tegmark (2023); Zou et al. (2023) show that the hidden representations $\mathbf{H}^{(l)}$ entail non-trivial information about the internal states of LLMs. These internal states are capable of being used to detect hallucinated generations from LLMs. One direct way is to train a factuality classifier with internal states as input Kadavath et al. (2022); Azaria & Mitchell (2023); Chen et al. (2023b); Zhang et al. (2024). Non-factual generation can also be detected as uncertainty of LLMs by internal state level consistency measuring among multiple generations Chen et al. (2023a).

These works potentially pave the way for improving adaptive RAG via examining the self-awareness from internal states. Since model decoding breaks down continuous internal states into discrete tokens, information loss during this process is inevitable. Compared with output-level self-awareness detection, internal states-level detection is more substantial and therefore better suited for adaptive RAG.

## 3 Self-Aware Knowledge Retrieval

As shown in Figure 2, SEAKR has three key components. 1) a *search engine* $S(\cdot)$, which returns ranked knowledge snippets according to the relevance to its input search query $\mathbf{qry}$. 2) a *large language model*, denoted as $\text{LLM}(\mathbf{c})$, which takes a context $\mathbf{c}$ as input, outputs a continuation to the context. Most importantly, (3) a *self-aware uncertainty estimator* $U(\mathbf{c})$, to quantify the uncertainty level of LLM to generate for input context $\mathbf{c}$.

For each input natural language question $\mathbf{q}$, SEAKR adopts a Chain-of-Thought (CoT) Wei et al. (2022) style iterative reasoning strategy. It maintains two buffers to collect retrieved knowledge $K = \{\mathbf{k}_i\}$ and generated rationales $R = \{\mathbf{r}_i\}$ during the iteration. During the $i^{\text{th}}$ iteration, SEAKR generates a rationale $\mathbf{r}_i$, before which it dynamically determines whether to augment the generation with external knowledge, *i.e.,* self-aware retrieval (§3.1). If SEAKR decides to invoke retrieval, it

adaptively selects knowledge $\mathbf{k}_i$ with self-aware re-ranking (§3.2). Finally, SEAKR integrates all previously gathered information, including $K$ and $R$, into the final answer, with self-aware reasoning (§3.3). At each stage, SEAKR utilizes the self-aware uncertainty estimator to measure the LLM uncertainty level from its internal states (§3.4).

### 3.1 SELF-AWARE RETRIEVAL

Self-aware retrieval relies on the self-aware uncertainty estimator $U(\cdot)$ to decide whether to use retrieved knowledge for rationale generation. In the following, we introduce our design to organize the input context, to generate the search query, and to generate the rationale.

**Input Context.** We first prepare the input context to prompt LLMs to generate one step of rationale without retrieval, and use $U(\cdot)$ to examine whether the LLM is uncertain so as to invoke retrieval accordingly. We organize $\mathbf{q}$ and historical rationales $R$ into the input context $\mathbf{c}_r$ using the following prompt template, we show details of the prompting template in Appendix D.1:

```
[In-Context Learning Examples]  Rationale [r1] Rationale [r2]
Rationale ...... For question:  [q] The Next Rationale is:
```

Here, placeholders are denoted in square brackets. Retrieval is triggered if the self-aware uncertainty exceeds an empirical threshold $U(\mathbf{c}_r) > \delta$.

**Query Generation.** To generate a query for the search engine, the LLM performs a pseudo-generation: $\mathbf{r}_s = \text{LLM}(\mathbf{c}_r)$. Tokens in $\mathbf{r}$ indicating high uncertainty due to their low probability are identified and removed from the pseudo-generated rationale to form the search query Jiang et al. (2023). We expect the retrieved knowledge to contain information that directly provides information to fill in the uncertain tokens in $\mathbf{r}_s$.

**Rationale Generation.** Finally, SEAKR generates rationale to proceed on answering the question $\mathbf{q}$. If retrieval is invoked, then knowledge snippets $\mathbf{k}$ are added to the current input context $\mathbf{c}_r$. Otherwise, the input context remains unchanged. The generated rationale $\mathbf{r} = \text{LLM}(\mathbf{c})$ is then appended to the rationale buffer.

### 3.2 SELF-AWARE RE-RANKING

Traditional RAG ranks the retrieved knowledge according to its relevance to the posed query. This approach overlooks how the retrieved knowledge aligns with the intrinsic knowledge of LLMs, potentially leading to performance degradation when the retrieved information contradicts the model's internal knowledge. Unlike existing methods, SEAKR prioritizes the *utility of the retrieved knowledge in reducing the LLM's self-aware uncertainty*. It selects the knowledge that most effectively reduces the LLM's uncertainty.

Specifically, SEAKR allows the search engine to retrieve multiple knowledge pieces. We preserve the top $N$ results and organize them along with previously generated rationales using the following template (Detailed in Appendix D.2):

```
[In-Context Learning Examples]  Rationale [r1]  Rationale ......
Knowledge Evidence: [k] For question: [q] The Next Rationale is:
```

As the search engine recalls top $N$ different knowledge snippets, SEAKR creates $N$ input contexts and evaluates their corresponding self-aware uncertainty from the LLM. The knowledge piece with the least uncertainty evaluated by $U(\cdot)$ is selected.

### 3.3 SELF-AWARE REASONING

The retrieval process within the SEAKR system halts under two conditions: 1) the LLM signals the end of generation with a prefatory statement, *"So the final answer is"*, terminating the iteration; 2) the retrieval activity reaches the maximum limit.

To effectively synthesize all previously retrieved knowledge, SEAKR employs two distinct reasoning strategies: *1) Reasoning with generated rationales R.* This approach prompts the LLM to

directly generate the final answer. It puts the instruction *"So the final answer is"* right after the last generated rationale. *2) Reasoning with retrieved knowledge $K$*. This strategy involves concatenating all re-ranked retrieved knowledge, which is then prepended to the question to serve as a reference context. SEAKR then requires the LLM to engage in CoT reasoning based on this augmented textual context. We show detailed prompting templates in Appendix D.3. The final answer is generated using the strategy that promotes the lowest level of uncertainty evaluated by $U(\cdot)$ between the answers generated with these two strategies.

### 3.4 SELF-AWARE UNCERTAINTY ESTIMATOR

For input context $\mathbf{c} = \langle\text{IN}\rangle_1 \cdots \langle\text{IN}\rangle_n$ with $n$ tokens, LLM works as a probabilistic distribution conditioned on the input context. To generate, it outputs $\mathbf{o}$ with $m$ tokens ending with an $\langle\text{EOS}\rangle$ token: $\text{LLM}(\mathbf{c}) = \langle\text{OUT}\rangle_1 \cdots \langle\text{OUT}\rangle_m \langle\text{EOS}\rangle$. We aim to extract how certain LLMs are that $\mathbf{o}$ is a correct continuation for $\mathbf{c}$. To this end, we follow INSIDE Chen et al. (2023a) and measure the uncertainty in the hidden space of the $\langle\text{EOS}\rangle$ token.

Specifically, for an input context $\mathbf{c}$, we first sample generation and preserve the hidden representation for its $\langle\text{EOS}\rangle$ token, denoted as $\mathbf{H}_{\langle\text{EOS}\rangle}^{(l)}$. As $\langle\text{EOS}\rangle$ attends to all previous tokens, it compresses information on both the output and the input. Then, we treat $\mathbf{H}_{\langle\text{EOS}\rangle}^{(l)}$ as a random variable, and sample $k$ different generations from the LLM for the same input context, whose $\mathbf{H}_{\langle\text{EOS}\rangle}^{(l)}$ are subsequently used to compute their Gram matrix Horn & Johnson (2012), which measures the correlation among each pair of representations. Finally, the uncertainty of the LLM is evaluated as the determinant of the regularized Gram matrix, a score of the consistency among a set of representations.

SEAKR uses the regularized Gram determinant as the self-aware uncertainty score for two reasons. 1) Pre-trained LLMs are proved to be well-calibrated probabilistic models, which behave less consistently when producing incorrect contents Kadavath et al. (2022); Zhu et al. (2023). 2) The Gram determinant examines the consistency on the internal state level, free from the influence of natural language where the same semantics can be expressed differently Qi et al. (2022).

## 4 EXPERIMENTS

In this section, we conduct experiments to compare SEAKR with baseline RAG methods that are commonly used on question answering (QA) tasks.

### 4.1 EXPERIMENT SETUP

We introduce the benchmark datasets used in the experiments and the baseline methods. We also describe key implementation details for SEAKR.

#### 4.1.1 BENCHMARK DATASETS

We use knowledge-intensive QA tasks, including both complex QA and simple QA.

**Complex QA** requires the model to perform multi-hop reasoning to answer the questions. Each question also needs multiple supporting knowledge. Specifically, for complex QA tasks, we test on 2WikiMultiHopQA (2Wiki, Ho et al., 2020), HotpotQA (HPQA, Yang et al., 2018), and the answerable subset of IIRC Ferguson et al. (2020).

**Simple QA** does not require multi-hop reasoning. These questions focus more on evaluating accurate knowledge acquisition. We use NaturalQuestions (NQ Kwiatkowski et al., 2019), TriviaQA (Joshi et al., 2017), and SQuAD (Rajpurkar et al., 2016) in the experiments.

We use them in the open-domain QA setting, where documents for machine reading comprehension are discarded. For dataset splitting, SEAKR is tuning-free and thus does not need a training set. We use a sampled subset from NQ's training split to search for hyper-parameters, which are adopted by all other datasets. We follow IRCoT Trivedi et al. (2022) to use their official development set and DPR Karpukhin et al. (2020) for simple QA.

### 4.1.2 BASELINES

We mainly compare SEAKR with representative RAG models, which include:

**Non-adaptive RAG**-based methods. • **Chain-of-Thought (CoT)** Wei et al. (2022) prompts an LLM to answer questions with multi-step explanations. We implement CoT with similar prompts as SEAKR by removing the retrieval-related instructions. • **IRCoT** Trivedi et al. (2022) interweaves CoT reasoning with retrieval augmented generation strategy. IRCoT retrieves for every reasoning step by default and integrates the top-ranked knowledge.

**Adaptive RAG**-based methods. • **Self-RAG** Asai et al. (2023) fine-tunes the LLM to generate a special token to indicate whether they need retrieval. The LLM is also trained to criticize the retrieved knowledge. The training data is generated by GPT-4 Achiam et al. (2023) with seed questions from NaturalQuestions Kwiatkowski et al. (2019). • **FLARE** Jiang et al. (2023) triggers retrieval when the LLM generates tokens with low probability. If so, it retrieves knowledge and regenerates the answer. The original FLARE does not support complex QA. We re-implement FLARE with IRCoT strategy to support evaluation on complex QA. • **DRAGIN** Su et al. (2024) decides to retrieve when low-probability tokens are generated and reformulates the query based on attention weights.

### 4.1.3 IMPLEMENTATION AND VARIABLE CONTROL

To implement SEAKR, we use LLaMA-2-chat with 7 billion parameters as the backbone LLM. The search engine is implemented with BM25 Robertson et al. (2009) algorithm using Elastic Search. Following DRAGIN Su et al. (2024), we use the English version Wikipedia dumped on December 20, 2018 as the external knowledge source. For simple QA, which does not require multiple knowledge evidence, we constrain the search time to $1$. These choices and constraints are also applied to all our baseline methods for fair comparison.

For hyper-parameters, we empirically set the number of knowledge recalled by the search engine to $N = 3$. We sample the hidden representation for $\langle \text{EOS} \rangle$ for $k = 20$ times, and implement with vLLM Kwon et al. (2023) for parallel inference. The self-aware uncertainty threshold $\delta$ is searched on with the development set. We 10 examples for in-context learning. The internal states are extracted from the middle layer of the LLM, *i.e.,* $l = \frac{L}{2}$, where $L$ is the total layer number.

### 4.2 EXPERIMENT RESULTS

We conduct experiments and report the results in F1 measure and exact match (EM) score.

### 4.2.1 RESULTS ON COMPLEX QA

Table 1 shows experiment results on complex QA tasks. SEAKR achieves 36.0%, 39.7%, and 23.5% F1 scores on 2WikiMultiHop, HotpotQA, and IIRC, which outperforms the best baselines by 6.0%, 5.5%, and 0.6%, respectively. These results indicate that self-aware knowledge retrieval strategy is beneficial for solving complex questions. It is worth noting that IIRC is especially challenging as it requires many numerical reasoning steps, which is extremely difficult for LLMs with 7B parameters. As SEAKR does not optimize the numerical reasoning capability, the performance gain on IIRC is less obvious than on 2Wiki and HPQA.

Table 1: Experiment results on complex QA datasets. Shown in percentage (%).

| Models | 2Wiki | | HPQA | | IIRC | |
|---|---|---|---|---|---|---|
| | EM | F1 | EM | F1 | EM | F1 |
| CoT | 14.6 | 22.3 | 18.4 | 27.5 | 13.9 | 17.3 |
| IR-CoT | 18.9 | 26.5 | 21.4 | 30.4 | 17.8 | 21.6 |
| Self-RAG | 4.6 | 19.6 | 6.8 | 17.5 | 0.9 | 5.7 |
| FLARE | 14.3 | 21.3 | 14.9 | 22.1 | 13.6 | 16.4 |
| DRAGIN | 22.4 | 30.0 | 23.7 | 34.2 | 19.1 | 22.9 |
| SEAKR | **30.2** | **36.0** | **27.9** | **39.7** | **19.5** | **23.5** |

For detailed analysis, we can see from the table that CoT reasoning, even without retrieval augmentation, can still solve a non-trivial amount of complex questions, reaching even 22.3%, 27.5%, and 17.3% F1 measures on the three datasets. This owns to questions that fully fall into the knowledge boundary of existing language models. As CoT utilizes similar reasoning prompts as SEAKR, with differences only in their retrieval-related instructions, the performance gap between CoT and SEAKR mainly lies in SEAKR's awareness of

its knowledge insufficiency to answer the question. At the opposite extreme, IRCoT retrieves in every reasoning step, also lagging behind SEAKR. This observation testifies to our hypothesis that adaptively determining when to retrieve is necessary.

Compared with the only fine-tuning based adaptive RAG method—Self-RAG, we can see that Self-RAG achieves less satisfactory results. This is mainly caused by the distribution of its fine-tuning data, which is generated by GPT-4 Achiam et al. (2023) with demonstrations from NaturalQuestions, a simple QA dataset. The distribution shift from simple QA to complex QA largely undermines LLMs' capacity to perform self-aware RAG. In contrast, SEAKR, as a tuning-free adaptive RAG method, achieves even better results. This shows that by exploring the intrinsic self-awareness of LLMs better generalizes to different QA tasks.

SEAKR outperforms FLARE and DRAGIN by a large margin. The most salient differences between SEAKR and FLARE / DRAGIN are two folds: 1) SEAKR determines the retrieval via self-aware uncertainty, while FLARE and DRAGIN superficially rely on output probability; 2) SEAKR is augmented with adaptive integration strategies, *i.e.,* self-aware re-ranking and self-aware reasoning, while FLARE and DRAGIN neglect this part. This performance gain is mainly due to these two improvements. We will conduct ablation study (§5.1) and case study (§5.4) to verify these reasons.

### 4.2.2 RESULTS ON SIMPLE QA

Table 2 shows results on simple QA tasks. SEAKR achieves the best performance among baselines on TriviaQA and SQuAD, at $63.1\%$ and $36.5\%$ F1 measure, On NaturalQuestions, SEAKR demonstrates comparable performance with tuning-free baseline FLARE, while lagging behind Self-RAG, which is fine-tuned to determine when to retrieve on GPT-4 generated NaturalQuestions-style data. The experiment results show that SEAKR is effective for questions that do not require reasoning.

We note that the performance gap between SEAKR and baselines in simple QA is less obvious than in complex QA datasets, especially on NQ and TriviaQA. This is because knowledge integration for simple questions is comprehended as a single machine reading comprehension step, which demands less on the capability to integrate external knowledge.

Table 2: Experiment results on simple QA datasets in percentage (%). Self-Rag is fine-tuned from LLaMA-2-chat (7B) with NQ style data. IR-CoT is not included as Simple QA do not require multiple retrieval.

| Model | NQ | | TriviaQA | | SQuAD | |
|---|---|---|---|---|---|---|
| | EM | F1 | EM | F1 | EM | F1 |
| CoT | 13.4 | 18.7 | 42.6 | 48.6 | 8.7 | 13.6 |
| Self-RAG | **32.3** | **40.2** | 21.2 | 37.9 | 5.1 | 18.3 |
| FLARE | 25.3 | 35.9 | 51.5 | 60.3 | 19.4 | 28.3 |
| DRAGIN | 23.2 | 33.2 | 54.0 | 62.3 | 18.7 | 28.7 |
| SEAKR | 25.6 | 35.5 | **54.4** | **63.1** | **27.1** | **36.5** |

## 5 ANALYSIS

We follow conventions Trivedi et al. (2022); Jiang et al. (2023) to sample 500 questions from each dataset to reduce the cost in analysis experiments.

### 5.1 ABLATION STUDY

We conduct the ablation study to verify the effectiveness of each component in SEAKR and explore alternative implementations. We show our ablation study results in Table 3.

**Ablating Self-aware Uncertainty Estimator.** We explore multiple ways to extract self-aware uncertainty from the LLM. The prompting-based method asks the LLM *"do I have sufficient knowledge to solve the question?"* and judges its uncertainty from the output directly. The perplexity-based method estimates the self-aware uncertainty based on the perplexity of the pseudo-generated contents. Multi-Perplexity estimates the uncertainty by averaging the perplexity of multiple generations, where we generate 20 times. Length normalized entropy (LN-Entropy, Malinin & Gales, 2020) is another uncertainty estimator for auto-regressive language models. Energy score calculates the uncertainty in the logit space, which is originally proposed to detect out-of-distribution samples Liu et al. (2020).

**Ablating Self-aware Retrieval.** To ablate the self-aware retrieval, we retrieve knowledge for each generation step, without dynamically determining when to retrieve ($-$ S.A. Retrieval). We can see that experiments on both the complex QA and simple QA degrade, indicating that when the LLM does not supplement knowledge, retrieved information indeed misleads LLM into generating incorrect information. Thus, it is necessary to determine when to retrieve dynamically to avoid such interference.

**Ablating Self-aware Re-ranking.** We ablate the self-aware re-ranking by choosing the first knowledge from the search engine, without utilizing the self-aware uncertainty score to select knowledge ($-$ S.A. Re-rank). From Table 3 we see that discarding self-aware re-ranking undermines the performance of SEAKR. This is because the self-aware re-ranking functions by de-noising retrieved knowledge, which integrates external knowledge resources more flexibly.

Table 3: Ablations study results. S.A. is abbreviate for self-aware. SEAKR performs differently from Table 1 and Table 2 due to dataset sampling. Self-aware reasoning only applies to complex QA as simple QA does not require multiple retrieval.

| Models | 2Wiki | | HPQA | | NQ | |
|---|---|---|---|---|---|---|
| | EM | F1 | EM | F1 | EM | F1 |
| SEAKR | 31.4 | 37.8 | 27.4 | 38.1 | 25.6 | 36.1 |
| *Ablating Self-aware Uncertainty Estimator* | | | | | | |
| Prompt | 27.0 | 33.9 | 26.5 | 37.3 | 23.8 | 34.2 |
| Perplexity | 29.0 | 35.2 | 26.6 | 36.9 | 23.0 | 33.4 |
| LN-Entropy | 30.0 | 36.0 | 26.2 | 37.5 | 24.8 | 34.8 |
| Energy | 26.8 | 33.2 | 22.2 | 31.7 | 22.8 | 32.3 |
| *Ablating Self-aware Retrieval* | | | | | | |
| $-$ S.A. Retrieval | 29.0 | 35.7 | 26.8 | 37.6 | 25.4 | 35.8 |
| *Ablating Self-aware Re-Ranking* | | | | | | |
| $-$ S.A. Re-rank | 29.2 | 35.0 | 26.2 | 36.6 | 24.8 | 35.0 |
| *Ablating Self-aware Reasoning* | | | | | | |
| Rationales-only | 29.4 | 35.9 | 26.6 | 36.3 | / | / |
| Knowledge-only | 30.4 | 37.0 | 27.6 | 37.2 | / | / |

Comparing the effect between removing self-aware retrieval and self-aware re-ranking, we observe that ablating self-aware re-ranking reduces the performance of SEAKR more than removing self-aware retrieval. This indicates the crucial aspect of designing effective knowledge integration method in adaptive RAG.

**Ablating Self-aware Reasoning.** We ablate self-aware reasoning by choosing two default reasoning strategies without adaptive choosing. Rationale-only prompts the LLM to generate the final answer directly after the last generated rationale. Knowledge-only concatenates the question with all previously selected knowledge $K$ to require the LLM to synthesize the final answer with CoT reasoning. Both the two strategies perform inferior to the original SEAKR. We interpret the results from two different angles. (1) The self-aware reasoning integrates all previously retrieved knowledge more effectively. (2) The self-aware reasoning functions as ensemble learning. Thus, self-aware reasoning exceeds each individual strategy Murphy (2012).

## 5.2 BACKBONE LLMS

To examine whether SEAKR scales to more powerful LLMs, we substitute the backbone LLM with LLaMA-3 with 8 billion parameters, which is pre-trained with more than $10\times$ FLOPS than LLaMA-2 (7B). We also examine the effectiveness of alignment tuning of the backbone LLM, and compare with the chat version of LLaMA-2 and instruct version of LLaMA-3.

Table 4 shows the comparisons. We find that SEAKR benefits from stronger backbone LLMs (*i.e.,* LLaMA-3), indicating that the effectiveness of SEAKR scales positively with the sophistication and ca-

Table 4: Experiments for backbone LLMs.

| Models | 2Wiki | | HPQA | | NQ | |
|---|---|---|---|---|---|---|
| | EM | F1 | EM | F1 | EM | F1 |
| *LLaMA-2 with 7B Parameters* | | | | | | |
| Base Version | 20.4 | 26.9 | 22.0 | 30.1 | 15.0 | 20.8 |
| Chat Version | 31.4 | 37.8 | 27.4 | 38.1 | 25.6 | 36.1 |
| *LLaMA-3 with 8B Parameters* | | | | | | |
| Base Version | 38.4 | 44.7 | 29.2 | 39.2 | 25.0 | 33.9 |
| Instruct Version | 40.6 | 48.1 | 36.0 | 47.7 | 31.0 | 43.0 |

pacity of the underlying language models. Another observation is that backbone LLMs with align-

Table 5: Case study. #Search denotes the knowledge rank given by the search engine. #$U(\mathbf{c})$ is the ranking according to self-aware uncertainty. SEAKR answers the question with two iterations and here we show the overall process of the second iteration. SEAKR first performs pseudo-generation, which results in high uncertainty $U(\mathbf{c}) = -4.4$ and triggers retrieval. The first returned knowledge from the search engine is relevant to the *Alejandro Jodorowsky* with certains dates, but does not help in answering the question. In contrast, the second retrieved knowledge reduces the self-aware uncertainty most, and indeed contains the critical information. We also notice that the third retrieved knowledge has overlapped information with the second one, which also result in a relatively low uncertainty score.

| Question (HPQA): | Who lived longer, Alejandro Jodorowsky or Philip Saville? | | Ground-Truth Answer: | Alejandro Jodorowsky |
|---|---|---|---|---|
| **Knowledge Buffer:** | . . . Philip Saville (sometimes credited as Philip Savile, 28 October 1930 – 22 December 2016) was a British television and film director, screenwriter and former actor . . . | | | |
| **Rationale Buffer:** | Philip Saville was born on 28 October 1930 and passed away on 22 December 2016. | | | |
| **Pseudo-Generation:** | Alejandro Jodorowsky was born on **7 July** 1929. | | **Self-aware Uncertainty:** | $U(\mathbf{c}) = -4.4, U(\mathbf{c}) > \delta$ |

| #Search | #$U(\mathbf{c})$ | $U(\mathbf{c})$ | Retrieved Knowledge Ranked by Search Engine $S(\mathbf{qry})$ |
|---|---|---|---|
| 1 | 3 | −4.37 | . . . interview with "The Guardian" newspaper in November 2009, however, Jodorowsky revealed that he was unable to find the funds to make "King Shot", and instead would be entering preparations on "Sons of El . . . |
| 2 | 1 | −4.91 | Alejandro Jodorowsky Prullansky (born **17 February 1929**) is a Chilean-French filmmaker . . . |
| 3 | 2 | −4.88 | . . . Alejandro Jodorowsky Prullansky (born **17 February 1929**) is a Chilean-French filmmaker. Since . . . |

ment tuning achieve higher performance. This is because of their better instruction-following capability to solve complex tasks.

## 5.3 HYPER-PARAMETER SEARCH

We search hyper-parameters for the knowledge recall size $N$, the dimension of the Gram determinant $k$, and the uncertainty threshold $\delta$ on a sample of training set of NQ. The exploration results are shown in Figure 3. The best number of generations to compute the Gram determinant $k$ falls into the interval $[10 - 25]$. The most indicative internal state is extracted from the middle layer, at $l = 16$. To determine the condition for the LLM to demand retrieval, we use $\delta > -6$ as the cut point to trigger retrieval, under which condition less than $80\%$ questions cannot be answered correctly. Our implementation for SEAKR is in line with these results.

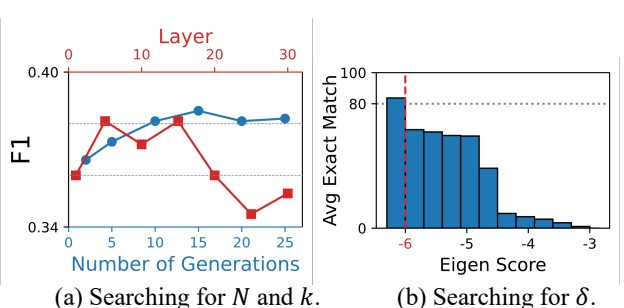

(a) Searching for $N$ and $k$.  (b) Searching for $\delta$.

Figure 3: Hyper-parameter search results.

## 5.4 CASE STUDY

In Table 5, we show an example on how SEAKR answers a question from HotpotQA. The main observations are two folds— 1) SEAKR accurately identifies its knowledge insufficiency. We observe this from its false pseudo-generation, where the LLM reckons the birthday of *Alejandro Jodorowsky* as *7 July 1929*. Luckily, SEAKR indeed gives a relatively high self-aware uncertainty estimation, and invokes retrieval timely. 2) SEAKR effectively integrates retrieved knowledge. We observe that the top-ranked knowledge from the search engine does not help with answering the question, while the knowledge that reduces the self-aware uncertainty most contains the information for the following step of reasoning. We show more cases in Appendix C to illustrate the effectiveness of each component in SEAKR.

## 6 CONCLUSION

In this paper, we propose self-aware knowledge retrieval (SEAKR) to perform adaptive RAG. SEAKR extracts self-aware uncertainty of LLMs from their internal states, and uses this as an indicator to invoke knowledge retrieval and dynamically integrate retrieved knowledge. Experiments on both complex QA and simple QA tasks show that SEAKR outperforms existing adaptive baselines. It is intriguing to further explore whether we can further enhance the uncertainty awareness via model training in the future.

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

## A    LIMITATIONS

We discuss the limitations of SEAKR.

**(1) Scope of Usage.** As SEAKR requires access to the internal state of LLMs, this limits the usability of SEAKR to open-sourced LLMs. However, the most powerful and widely adopted LLMs are still preserved by commercial companies, such as GPT series model. We still need to explore new ways to estimate the self-aware uncertainty from the output of the language model, rather than their internal states.

**(2) Task Coverage.** We mainly evaluate SEAKR on short-form question answering tasks, neglecting a broad spectrum of natural language processing tasks, such as long-form question answering, creative writing, etc.

**(3) Computation Issues.** To compute Gram determinant, SEAKR requires the backbone to conduct 20 pseudo-generations, which is computationally costly. We explore the engineering trick to mitigate this issue—by deploying the backbone LLM with vLLM Kwon et al. (2023), which implements paged attention to support parallel inference in a single batch. Thus, the latency of 20 pseudo-generation is roughly the same as a single pseudo-generation. All the experiments can be held on a single NVidia 3090 GPU with 24GiB GRAM.

**(4) Model Scaling.** Due to our limited computation resources, we are not able to deploy LLMs larger than one with 8 billion parameters. As recent evidences suggest that model scaling is more closely related to training FLOPS, rather than model scale. We thus compare between LLaMA-2 (7B) and LLaMA-3 (8B) to verify whether SEAKR is scalable to more powerful LLMs. This is because although they have similar parameter scales, LLaMA-3 is trained on $10\times$ more corpora, and thus $10\times$ more FLOPS than LLaMA-2.

**(5) Information Retrieval.** The authors would like to mention that, with the development of information retrieval technology, the second part of SEAKR (*i.e.,* Self-aware Re-ranking) could be surpassed by advanced IR methods, in the future.

## B    BROAD IMPACT

We discuss the ethical considerations and broader impact of SEAKR.

**(1) Intended Usage.** SEAKR falls into the category of retrieval augmented generation, which is intended to increase the factual correctness of LLMs. Thus, the intention of our work is to improve the trustworthiness of LLM.

**(2) Potential Misuse.** However, for detailed technology we adopted, it can be misused to create misleading information. For example, the self-aware uncertainty estimator can be used as an adversarial signal for model training, which could make models better at deceiving humans with uncertain information. Another issue is the increased integration of LLM and IR systems, which may be used to automate cyber manhunt.

**(3) Risk Control.** SEAKR is developed upon open-sourced LLMs. We will also release our code. We hope that transparency helps to monitor and prevent its mis-usage.

**(4) Intellectual Artifacts.** We cite the creator of our used intellectual artifacts. Specifically, we use 6 question answering benchmark dataset in this paper, they are 2WikiMultiHopQA Ho et al. (2020), HotpotQA Yang et al. (2018), IIRC Ferguson et al. (2020), NaturalQuestion Kwiatkowski et al. (2019), TriviaQA Joshi et al. (2017), and SQuAD Rajpurkar et al. (2016). We would also like to acknowledge creators of Self-RAG Asai et al. (2023), FLARE Jiang et al. (2023), and DRAGIN Su et al. (2024) for sharing their codebases, which are used to reproduce their methods, along with IRCoT. All the used intellectual artifacts' license allows for academic usage.

## C    CASE STUDY

### C.1    CASE STUDY FOR SELF-AWARE RETRIEVAL

We present additional examples for self-aware retrieval in Table 7.

Table 6: Dataset Statistics.

|  | Complex QA | | | Simple QA | | |
|---|---|---|---|---|---|---|
|  | TwoWiki | HotpotQA | IIRC | NQ | TriviaQA | SQuAD |
| #Examples | $12,576$ | $7,405$ | $954$ | $3,610$ | $11,313$ | $7,357$ |

In each step, SEAKR evaluates the uncertainty of the pseudo-generation and determines whether to retrieve external knowledge based on the predefined threshold. Three cases are presented: **Case #1**, where the generation fails to meet the predefined threshold and retrieval is triggered; **Case #2 & #3**, where the model correctly and confidently generates an output, bypassing potentially redundant retrieval; **Case #3** also shows that SEAKR successfully performs self-aware retrieval amidst multi-step reasoning, where the knowledge buffer and the rationale buffer are not empty.

### C.2 CASE STUDY FOR SELF-AWARE RE-RANKING

We present additional examples for self-aware re-ranking in Table 8.

After the retrieval is invoked, SEAKR performs pairwise re-ranking and identifies the optimal passage for generating subsequent reasoning steps. Here, to determine what is the original parent company of *FastJet Tanzania*, three pieces of external knowledge (passages) are retrieved, where **Knowledge #1** and **Knowledge #3** present distractions—listing the current headquarters (*Dar es Salaam*) and the major shareholder (*Fastjet Plc*), while **Knowledge #2** contains critical information that it was founded as a subsidiary of a Kenya company. SEAKR's self-aware uncertainty gives an effective re-ranking and prioritizes **Knowledge #2**.

### C.3 CASE STUDY FOR SELF-AWARE REASONING

Table D.3 illustrates an additional example of self-aware reasoning.

In this case, SEAKR adaptively selects the optimal answer from two strategies: one generated from all rationales and the other from all knowledge. The initial rationale incorrectly asserts that *Stormbreaker* is a fantasy film, misleading the reasoning afterward and exhibiting poor uncertainty scores. In contrast, when reasoning from all evidence passages, SEAKR regenerates each step from scratch, utilizing more informative knowledge retrieved in the second step (The *Spiderwick Chronicles* is the fantasy film that has *Sarah Bolger* in it). It also results in a better uncertainty score, at $-6.20$, than the average rationale score $-4.73$.

Table 7: Additional examples for self-aware retrieval.

| | *Case #1* |
|---|---|
| **Question (HPQA):** | In what city is the company that Fastjet Tanzania was originally founded as a part of prior to rebranding based? |
| **Ground-Truth Answer:** | Nairobi, Kenya |
| **Pseudo-Generation:** | FastJet Tanzania was originally founded as a part of **the company Fastjet plc, which was based in London, United Kingdom**. |
| **Gold-Fact:** | Fastjet Airlines Limited (Tanzania), also known as Fastjet Tanzania, was founded in 2011 as Fly540 Tanzania. |
| **Self-aware Uncertainty:** | $U(\mathbf{c}) = -4.84, U(\mathbf{c}) > \delta$, Need to retrieve ✗ |
| | *Case #2* |
| **Question (HPQA):** | The Argentine National Anthem was adopted 3 years after which event that led to the removal of Viceroy Baltasar Hildalgo de Cisneros? |
| **Ground-Truth Answer:** | May Revolution |
| **Pseudo-Generation:** | The Argentine National Anthem was adopted in 1813 . |
| **Gold-Fact:** | The National Anthem of Argentina, or the Himno Nacional Argentino as it is known to its citizens, was adopted on May 11, 1813. |
| **Self-aware Uncertainty:** | $U(\mathbf{c}) = -6.11, U(\mathbf{c}) < \delta$, No need to retrieve ✓ |
| | *Case #3* |
| **Question (HPQA):** | Stephen Smith appears on ESPN First Take alongside which HBO boxing commentator? |
| **Ground-Truth Answer:** | Max Kellerman |
| **Knowledge Buffer:** | Stephen A. Smith Stephen Anthony Smith (born October 14, 1967) is an American sports television personality, sports radio host, sports journalist, and actor. Smith is a commentator on "ESPN First Take", where he appears with Max Kellerman and Molly Qerim. He also makes frequent appearances as an NBA analyst on "SportsCenter". He also is an NBA analyst for ESPN on "NBA Countdown" and NBA broadcasts on ESPN. Smith formerly hosted "The Stephen A. Smith and Ryan Ruocco Show" on ESPN Radio New York 98.7 FM. He now hosts "The Stephen A. Smith Show" on the Chris Russo sports radio station: |
| **Rationale Buffer:** | Stephen Smith appears on ESPN First Take alongside Max Kellerman and Molly Qerim |
| **Pseudo-Generation:** | Max Kellerman **is an HBO boxing commentator**. |
| **Gold-Fact:** | Max Kellerman (born August 6, 1973) is an American sports television personality and boxing commentator |
| **Self-aware Uncertainty:** | $U(\mathbf{c}) = -6.03, U(\mathbf{c}) < \delta$, No need to retrieve ✓ |

# D PROMPT TEMPLATES

## D.1 SELF-AWARE RETRIEVAL

At the beginning of each iteration of reasoning, SEAKR executes and evaluates a pseudo-generation. We set the stop token to a period (.) to limit the generation to the next single step.

---

**Self-aware Retrieval**

[ICL Examples]

**Question:** [INPUT QUESTION]
**Answer:**

---

## D.2  SELF-AWARE RE-RANKING

When the uncertainty score of direct generation fails to meet the threshold, SEAKR retrieves and re-rank a pseudo-generation in a pair-wise manner. We also set the stop token to a period (.).

---

**Self-aware Re-ranking**

[ICL Examples]

**Context:**
**[1]**. [Retrieved Doc 1]
Answer in the same format as before.
**Question:** [INPUT QUESTION]
**Answer:**

---

## D.3  SELF-AWARE REASONING

In the final stage, SEAKR selects the optimal response either from the rationales or directly from the knowledge. For the rationales, we extract the answer following the phrase "So the answer is" in the last rationale. For the knowledge group, we perform a full CoT reasoning using all retrieved passages. The stop token in both groups is the newline character \n.

---

**Self-aware Reasoning with retrieved knowledge**

[ICL Examples]

**Context:**
**[1]**. [Retrieved Doc 1]
**[2]**. [Retrieved Doc 1]
**[3]**. [Retrieved Doc 1]
Answer in the same format as before.
**Question:** [INPUT QUESTION]
**Answer:**

---

**Self-aware Reasoning with generated rationales**

[ICL Examples]

**Question:** [INPUT QUESTION]
**Answer:**
**[Step 1]**.
**[Step 2]**.
**[Step 3]**.
So the answer is

---

## D.4  IN CONTEXT LEARNING EXAMPLES

We use the same in-context-learning examples for simple QA datasets (Fig. 4) and different examples for each multihop QA dataset followed by IRCoTTrivedi et al. (2022): 2WikiMultiHopQA(Fig. 5), HotpotQA(Fig. 6), and IIRC (Fig. 7).

# E    DATASETS AND SETTINGS

Dataset statistics are summarized in Table 6. We conduct the hyperparameter search using $3,000$ samples from the training set of the Natural Questions dataset.

We modified the source code of vLLM 0.4.2 to compute uncertainty scores based on internal states. This implementation uses PyTorch 2.3.0. For the retrieval component, we employ Elasticsearch 7.17.9 to run a local retrieval service.

Table 8: Additional examples for self-aware re-ranking.

| | |
|---|---|
| **Question (HPQA):** | In what city is the company that Fastjet Tanzania was originally founded as a part of prior to rebranding based? |
| **Ground-Truth Answer:** | Nairobi, Kenya |
| **Failed Direct Output:** | FastJet Tanzania was originally founded as a part of **the company Fastjet plc, which was based in London, United Kingdom**. |
| **Gold-Fact:** | Fastjet Airlines Limited (Tanzania), also known as Fastjet Tanzania, was founded in 2011 as Fly540 Tanzania. Fly540, is a low-cost airline which commenced operations in 2006 and is based in Nairobi, Kenya. |

| | |
|---|---|
| **Query:** | FastJet Tanzania originally founded as part |

| *Knowledge #1* | |
|---|---|
| **Passage:** | Plc group accounts. Some information has been made available for the Tanzanian operation (as at year ending 31 December): Fastjet Tanzania maintains a head office in Samora Avenue, Dar es Salaam, Tanzania. As of 4 November 2017, Fastjet Tanzania serves the following destinations: Fastjet has signed an agreement with one of Africa's largest cargo operators, BidAir Cargo, to carry cargo on its fleet of Airbus A319s. Fastjet has sufficient capacity to accommodate the carrying of cargo on its Tanzanian routes The Fastjet Tanzania fleet includes the following aircraft as of June 2017: Fastjet Tanzania Fastjet Airlines Limited (Tanzania), also known |
| **Pseudo-Generation:** | Fastjet Tanzania was originally founded as a part of **prior to rebranding based in Dar es Salaam, Tanzania**. |
| **Self-aware Uncertainty:** | $U(\mathbf{c}) = -5.10$ ✗ |

| *Knowledge #2* | |
|---|---|
| **Passage:** | Fastjet Tanzania Fastjet Airlines Limited (Tanzania), also known as Fastjet Tanzania, is a low-cost airline that operates flights under the fastjet brand in Tanzania. The airline was founded in 2011 as "Fly540 Tanzania", but through the acquisition of Fly540 in 2012, it was rebranded as Fastjet Tanzania. It is based in Dar es Salaam. The airline carried more than 350,000 passengers in the first year of operations and sold one million seats by December 2014. Fastjet Tanzania was founded in 2011 as "Fly540 Tanzania", a subsidiary of Kenya-based Fly540. Using a Bombardier CRJ100 and a Dash 8-100, |
| **Pseudo-Generation:** | Fastjet Tanzania was originally founded as a part of **Fly540, which is based in Nairobi, Kenya**. |
| **Self-aware Uncertainty:** | $U(\mathbf{c}) = -5.828{\downarrow}$ ✓ |

| *Knowledge #3* | |
|---|---|
| **Passage:** | It currently (August 2015) has domestic routes operating linking Dar es Salaam with Mwanza, Kilimanjaro and Mbeya, and four international routes from Dar es Salaam to Johannesburg, Harare, Entebbe, Lilongwe and Lusaka. Fastjet Tanzania is 49% owned by Fastjet Plc; on 14 November 2014 it was announced that Fastjet Plc had entered into an agreement to sell an interest in fastjet Tanzania to Tanzanian investors. The issue of the shares brings the total Tanzanian legal and beneficial ownership of fastjet Tanzania to 51 |
| **Pseudo-Generation:** | Fastjet Tanzania was originally founded as a part of **prior to rebranding based in Dar es Salaam, Tanzania**. |
| **Self-aware Uncertainty:** | $U(\mathbf{c}) = -5.302$ ✗ |

| *Rerank Result* | |
|---|---|
| **Selected Knowledge:** | Knowledge #2 |
| **Generated Rationale:** | Fastjet Tanzania was originally founded as a part of **Fly540, which is based in Nairobi, Kenya**. |

Table 9: Additional examples for self-aware reasoning

| | |
|---|---|
| **Question (HPQA):** | What's the name of the fantasy film starring Sarah Bolger, featuring a New England family who discover magical creatures around their estate? |
| **Ground-Truth Answer:** | The Spiderwick Chronicles |
| **Rationale Buffer:** | The fantasy film starring Sarah Bolger is "Stormbreaker" |
| | It features a New England family who discover magical creatures around their estate. |
| | So the answer is Stormbreaker. |
| **Knowledge Buffer:** | Hard to Find" directed by Abner Pastoll. Filming completed in December 2017, with a release slated for 2018. In January 2011, Bolger was selected to be in photographer Kevin Abosch's project "The Face of Ireland" alongside other Irish celebrities including Sinéad O'Connor, Neil Jordan, and Pierce Brosnan. Sarah Bolger Sarah Bolger (born 28 February 1991) is an Irish actress. She has starred in the films "In America", "Stormbreaker", "The Spiderwick Chronicles" and "Emilie". She is also known for her role as Lady Mary Tudor in the TV series "The Tudors", for which she won an IFTA award, and for her |
| | The Spiderwick Chronicles (film) The Spiderwick Chronicles is a 2008 American fantasy adventure film based on the bestselling book series of the same name by Holly Black and Tony DiTerlizzi. It was directed by Mark Waters and stars Freddie Highmore, Sarah Bolger, Mary-Louise Parker, Martin Short, Nick Nolte, and Seth Rogen. Set in the Spiderwick Estate in New England, it follows the adventures of Jared Grace and his family as they discover a field guide to fairies while battling goblins, mole trolls, and other magical creatures. Produced by Nickelodeon Movies and distributed by Paramount Pictures, it was released on February |
| | ESRB. The Spiderwick Chronicles (film) The Spiderwick Chronicles is a 2008 American fantasy adventure film based on the bestselling book series of the same name by Holly Black and Tony DiTerlizzi. It was directed by Mark Waters and stars Freddie Highmore, Sarah Bolger, Mary-Louise Parker, Martin Short, Nick Nolte, and Seth Rogen. Set in the Spiderwick Estate in New England, it follows the adventures of Jared Grace and his family as they discover a field guide to fairies while battling goblins, mole trolls, and other magical creatures. Produced by Nickelodeon Movies and distributed by Paramount Pictures, it was released on |
| | *From Rationales* |
| **Answer:** | Stormbreaker. |
| **Self-aware Uncertainty:** | $((-5.25) + (-5.38) + (-3.56))/3 = -4.73$. |
| | *From Knowledge* |
| **Answer:** | The fantasy film starring Sarah Bolger is The Spiderwick Chronicles. So the answer is The Spiderwick Chronicles. |
| **Self-aware Uncertainty:** | $-6.20$.✓ |
| | *Final Answer* |
| | The Spiderwick Chronicles. |

**Examples for Simple QA**

**Question:** Nobody Loves You was written by John Lennon and released on what album that was issued by Apple Records, and was written, recorded, and released during his 18 month separation from Yoko Ono?
**Answer:** The album issued by Apple Records, and written, recorded, and released during John Lennon's 18 month separation from Yoko Ono is Walls and Bridges.
Nobody Loves You was written by John Lennon on Walls and Bridges album.
So the answer is Walls and Bridges.

**Question:** What is known as the Kingdom and has National Route 13 stretching towards its border?
**Answer:** Cambodia is officially known as the Kingdom of Cambodia.
National Route 13 stretches towards the border to Cambodia.
So the answer is Cambodia.

**Question:** Jeremy Theobald and Christopher Nolan share what profession?
**Answer:** Jeremy Theobald is an actor and producer.
Christopher Nolan is a director, producer, and screenwriter.
Therefore, they both share the profession of being a producer.
So the answer is producer.

**Question:** What film directed by Brian Patrick Butler was inspired by a film directed by F.W. Murnau?
**Answer:** Brian Patrick Butler directed the film The Phantom Hour.
The Phantom Hour was inspired by the films such as Nosferatu and The Cabinet of Dr. Caligari.
Of these, Nosferatu was directed by F.W. Murnau.
So the answer is The Phantom Hour.

**Question:** Vertical Limit stars which actor who also played astronaut Alan Shepard in 'The Right Stuff'?
**Answer:** The actor who played astronaut Alan Shepard in 'The Right Stuff' is Scott Glenn.
The movie Vertical Limit also starred Scott Glenn.
So the answer is Scott Glenn.

**Question:** Which car, produced by Ferrari from 1962 to 1964 for homologation into the FIA's Group 3 Grand Touring Car category inspired the Vandenbrink GTO?
**Answer:** The car produced by Ferrari from 1962 to 1964 for homologation into the FIA's Group 3 Grand Touring Car category is the Ferrari 250 GTO.
The Ferrari 250 GTO also inspired the Vandenbrink GTO's styling.
So the answer is Ferrari 250 GTO.

**Question:** The actor that stars as Joe Proctor on the series 'Power' also played a character on 'Entourage' that has what last name?
**Answer:** The actor that stars as Joe Proctor on the series 'Power' is Jerry Ferrara.
Jerry Ferrara also played a character on Entourage named Turtle Assante.
Thus, Turtle Assante's last name is Assante.
So the answer is Assante.

**Question:** In which country did this Australian who was detained in Guantanamo Bay detention camp and published 'Guantanamo: My Journey' receive para-military training?
**Answer:** The Australian who was detained in Guantanamo Bay detention camp and published 'Guantanamo: My Journey' is David Hicks.
David Hicks received his para-military training in Afghanistan.
So the answer is Afghanistan.

**Question:** Does The Border Surrender or Unsane have more members?
**Answer:** The Border Surrender band has following members: Keith Austin, Simon Shields, Johnny Manning and Mark Austin. That is, it has 4 members.
Unsane is a trio of 3 members.
Thus, The Border Surrender has more members.
So the answer is The Border Surrender.

Figure 4: Examples for Simple QA.

## 2WikiMultiHopQA

**Question:** Who was born first out of Martin Hodge and Ivania Martinich?
**Answer:** Martin Hodge was born on 4 February 1959.
Ivania Martinich was born on 25 July 1995.
Thus, 4 February 1959 is earlier than 25 July 1995 and Martin Hodge was born first.
So the answer is Martin Hodge.

**Question:** When did the director of film Hypocrite (Film) die?
**Answer:** The film Hypocrite was directed by Miguel Morayta.
Miguel Morayta died on 19 June 2013.
So the answer is 19 June 2013.

**Question:** Are both Kurram Garhi and Trojkrsti located in the same country?
**Answer:** Kurram Garhi is located in the country of Pakistan.
Trojkrsti is located in the country of Republic of Macedonia.
Thus, they are not in the same country.
So the answer is no.

**Question:** Do the director of film Coolie No. 1 (1995 Film) and the director of film The Sensational Trial have the same nationality?
**Answer:** Coolie No. 1 (1995 film) was directed by David Dhawan.
The Sensational Trial was directed by Karl Freund.
David Dhawan's nationality is Indian.
Karl Freund's nationality is German.
Thus, they do not have the same nationality.
So the answer is no.

**Question:** Who is Boraqchin (Wife Of Ögedei)'s father-in-law?
**Answer:** Boraqchin is married to Ögedei Khan.
Ögedei Khan's father is Genghis Khan.
Thus, Boraqchin's father-in-law is Genghis Khan.
So the answer is Genghis Khan.

**Question:** When did the director of film Laughter In Hell die?
**Answer:** The film Laughter In Hell was directed by Edward L. Cahn.
Edward L. Cahn died on August 25, 1963.
So the answer is August 25, 1963.

**Question:** Who is the grandchild of Krishna Shah (Nepalese Royal)?
**Answer:** Krishna Shah has a child named Rudra Shah.
Rudra Shah has a child named Prithvipati Shah.
Thus, Krishna Shah has a grandchild named Prithvipati Shah.
So the answer is Prithvipati Shah.

**Question:** Where did the director of film Maddalena (1954 Film) die?
**Answer:** The film Maddalena is directed by Augusto Genina.
Augusto Genina died in Rome.
So the answer is Rome.

**Question:** What is the cause of death of Grand Duke Alexei Alexandrovich Of Russia's mother?
**Answer:** The mother of Grand Duke Alexei Alexandrovich of Russia is Maria Alexandrovna.
Maria Alexandrovna died from tuberculosis.
So the answer is tuberculosis.

**Question:** Which film has the director died later, The Gal Who Took the West or Twenty Plus Two?
**Answer:** The mother of Grand Duke Alexei Alexandrovich of The film Twenty Plus Two was directed by Joseph M. Newman.
The Gal Who Took the West was directed by Frederick de Cordova.
Joseph M. Newman died on January 23, 2006.
Fred de Cordova died on September 15, 2001.
Thus, January 23, 2006 is later than September 15, 2001, and the person to die later from the two is Twenty Plus Two.
So the answer is Twenty Plus Two.

23

Figure 5: Examples for 2WikiMultiHopQA

### HotpotQA

**Question:** Jeremy Theobald and Christopher Nolan share what profession?
**Answer:** Jeremy Theobald is an actor and producer.
Christopher Nolan is a director, producer, and screenwriter.
Therefore, they both share the profession of being a producer.
So the answer is producer.

**Question:** What film directed by Brian Patrick Butler was inspired by a film directed by F.W. Murnau?
**Answer:** Brian Patrick Butler directed the film *The Phantom Hour*.
*The Phantom Hour* was inspired by the films such as *Nosferatu* and *The Cabinet of Dr. Caligari*.
Of these, *Nosferatu* was directed by F.W. Murnau.
So the answer is *The Phantom Hour*.

**Question:** How many episodes were in the South Korean television series in which Ryu Hye-young played Bo-ra?
**Answer:** The South Korean television series in which Ryu Hye-young played Bo-ra is *Reply 1988*.
The number of episodes *Reply 1988* has is 20.
So the answer is 20.

**Question:** Were Lonny and Allure both founded in the 1990s?
**Answer:** Lonny (magazine) was founded in 2009.
Allure (magazine) was founded in 1991.
Thus, of the two, only Allure was founded in the 1990s.
So the answer is no.

**Question:** Vertical Limit stars which actor who also played astronaut Alan Shepard in *The Right Stuff*?
**Answer:** The actor who played astronaut Alan Shepard in *The Right Stuff* is Scott Glenn.
The movie *Vertical Limit* also starred Scott Glenn.
So the answer is Scott Glenn.

**Question:** What was the 2014 population of the city where Lake Wales Medical Center is located?
**Answer:** Lake Wales Medical Center is located in the city of Lake Wales, Polk County, Florida.
The population of Lake Wales in 2014 was 15,140.
So the answer is 15,140.

**Question:** Who was born first? Jan de Bont or Raoul Walsh?
**Answer:** Jan de Bont was born on 22 October 1943.
Raoul Walsh was born on March 11, 1887.
Thus, Raoul Walsh was born first.
So the answer is Raoul Walsh.

**Question:** In what country was *Lost Gravity* manufactured?
**Answer:** The *Lost Gravity* (roller coaster) was manufactured by Mack Rides.
Mack Rides is a German company.
So the answer is Germany.

**Question:** Which of the following had a debut album entitled 'We Have an Emergency': Hot Hot Heat or The Operation M.D.?
**Answer:** The debut album of the band 'Hot Hot Heat' was 'Make Up the Breakdown'.
The debut album of the band 'The Operation M.D.' was 'We Have an Emergency'.
So the answer is The Operation M.D..

**Question:** How many awards did the 'A Girl Like Me' singer win at the American Music Awards of 2012?
**Answer:** The singer of 'A Girl Like Me' is Rihanna.
In the American Music Awards of 2012, Rihanna won one award.
So the answer is one.

**Question:** The actor that stars as Joe Proctor on the series 'Power' also played a character on 'Entourage' that has what last name?
**Answer:** The actor that stars as Joe Proctor on the series 'Power' is Jerry Ferrara.
Jerry Ferrara also played a character on Entourage named Turtle Assante.
Thus, Turtle Assante's last name is Assante.
So the answer is Assante.

---

**IIRC**

**Question:** What is the age difference between the kicker and the quarterback for the Chargers?
**Answer:** The kicker for the Chargers is Nate Kaeding.
The quarterback (QB) for the Chargers is Philip Rivers.
Nate Kaeding was born in the year 1982.
Philip Rivers was born in the year 1981.
Thus, the age difference between them is of 1 year.
So the answer is 1.

**Question:** How many years was the ship that took the battalion from New South Wales to Ceylon in service?
**Answer:** The ship that took the battalion from New South Wales to Ceylon is *General Hewitt*.
*General Hewitt* was launched in Calcutta in 1811.
*General Hewitt* was sold for a hulk or to be broken up in 1864.
So she served for a total of 1864 - 1811 = 53 years.
So the answer is 53.

**Question:** What year was the theatre that held the 2016 NFL Draft built?
**Answer:** The theatre that held the 2016 NFL Draft is Auditorium Theatre.
The Auditorium Theatre was built in 1889.
So the answer is 1889.

**Question:** How long had Milan been established by the year that Nava returned there as a reserve in the first team's defense?
**Answer:** Nava returned to Milan as a reserve in the first team's defense in the year 1990.
Milan had been established in the year 1899.
Thus, Milan had been established for 1990 - 1899 = 91 years when Nava returned to Milan as a reserve in the first team's defense.
So the answer is 91.

**Question:** When was the town Scott was born in founded?
**Answer:** Scott was born in the town of Cooksville, Illinois.
Cooksville was founded in the year 1882.
So the answer is 1882.

**Question:** In what country did Wright leave the French privateers?
**Answer:** Wright left the French privateers in Bluefield's river.
Bluefields is the capital of the South Caribbean Autonomous Region (RAAS) in the country of Nicaragua.
So the answer is Nicaragua.

**Question:** Who plays the A-Team character that Dr. Hibbert fashioned his hair after?
**Answer:** Dr. Hibbert fashioned his hair after Mr. T from *The A-Team*.
Mr. T's birthname is Lawrence Tureaud.
So the answer is Lawrence Tureaud.

**Question:** How many people attended the conference held near Berlin in January 1942?
**Answer:** The conference held near Berlin in January 1942 is the Wannsee Conference.
The Wannsee Conference was attended by 15 people.
So the answer is 15.

**Question:** When did the country Ottwalt went into exile in founded?
**Answer:** Ottwalt went into exile in the country of Denmark.
Denmark has been inhabited since around 12,500 BC.
So the answer is 12,500 BC.

**Question:** When was the J2 club Uki played for in 2001 founded?
**Answer:** The J2 club that Uki played for is Montedio Yamagata.
Montedio Yamagata was founded in 1984.
So the answer is 1984.

---

Figure 7: Examples for IIRC

