# OpenReview forum: "SeaKR: Self-aware Knowledge Retrieval for Adaptive Retrieval Augmented Generation"
_ICLR.cc/2025/Conference — Submitted to ICLR 2025_

### Official Review · Reviewer_1gAL · 2024-10-16

**Soundness:** 2
**Presentation:** 3
**Contribution:** 2
**Rating:** 5
**Confidence:** 4

**Summary:**

The paper introduces SEAKR, a novel adaptive RAG model designed to improve the performance of LLMs by reducing hallucinations. SEAKR enhances RAG by dynamically invoking knowledge retrieval based on LLMs' internal self-aware uncertainty. When the LLMs experience high uncertainty during generation, SEAKR retrieves external knowledge, re-ranks retrieved snippets to reduce uncertainty, and integrates the most relevant information. This model also adapts reasoning strategies for complex tasks requiring multiple retrievals.

**Strengths:**

The paper proposes using the internal self-awareness of LLMs to enhance the effectiveness of adaptive RAG. Experimental results show that the proposed method significantly improves the accuracy of LLMs on multiple QA datasets, outperforming existing methods. The structure of the paper is very clear and easy to understand.

**Weaknesses:**

1. Some statements regarding the contributions in this paper are flawed. For example, the paper claims that existing adaptive RAG methods determine whether retrieval is needed solely based on the model’s output, which is clearly incorrect. Papers like Self-RAG[1] use the model’s in-context learning ability to judge whether additional knowledge retrieval is necessary, and during this process, the model naturally considers input prompts, including questions and retrieved knowledge. Additionally, many adaptive RAG methods re-rank retrieved knowledge during the integration process. For example, Self-RAG labels knowledge as "is_supportive," "is_related," and "is_useful." Similar work includes RankRAG[2], RagVL[3]. Therefore, the paper's claim that existing methods "neglecting to re-rank multiple retrieved knowledge and optimize the reasoning paths" is biased.

2. The method proposed in this paper mainly consists of three processes: retrieval, re-ranking, and reasoning. The overall pipeline design is not novel, and the innovation of this paper lies primarily in using the model's self-aware uncertainty to evaluate the output of each process to decide when to retrieve, how to re-rank the retrieved knowledge, and whether to use the model's generated rationale or the retrieved information during reasoning. However, based on the experimental results in Table 3, using the proposed Self-aware Uncertainty Estimator does not provide a particularly significant improvement compared to using LN-Entropy, perplexity, or prompt to determine the LLM's uncertainty. Therefore, the innovation of this paper is limited, and the improvements in experiments may be more attributable to the effectiveness of the pipeline itself, while the innovation in the pipeline design is trivial.

3. Computing self-aware uncertainty seems to be a very time-consuming and computationally expensive process, yet the paper does not discuss the time and computational cost of the entire reasoning process. On the other hand, although this method is tuning-free, it can only be applied to local models. For closed-source LLMs, since their hidden layers are inaccessible, this method is not applicable, which weakens the advantage of this method in being tuning-free and affects its practical value.

4. The experiment uses the F1 measure as an evaluation method, but in practice, when calculating F1, the model's output is matched word-by-word against the ground truth in an N-to-N fashion. This is not the standard F1 score evaluation method and does not reflect the method's retrieval performance. Moreover, this aspect was not clarified in the experimental design.

[1] Asai, Akari, et al. "Self-rag: Learning to retrieve, generate, and critique through self-reflection." arXiv preprint arXiv:2310.11511 (2023).
[2] Yu, Yue, et al. "RankRAG: Unifying Context Ranking with Retrieval-Augmented Generation in LLMs." arXiv preprint arXiv:2407.02485 (2024).
[3] Chen, Zhanpeng, et al. "MLLM Is a Strong Reranker: Advancing Multimodal Retrieval-augmented Generation via Knowledge-enhanced Reranking and Noise-injected Training." arXiv preprint arXiv:2407.21439 (2024).

**Questions:**

1. How long is the inference time required for the entire pipeline? Compared to the Prompt method, how much additional computation is needed when using the self-aware uncertainty estimator proposed in this paper? Is it applicable to larger LLMs?

2. Since the proposed self-aware uncertainty estimator can only be applied to local models, could the LLM be fine-tuned with an additonal adapter to better calculate uncertainty?

3. When determining whether retrieval is necessary, could a simpler and more effective approach be used, like Chain of Knowledge[4], where self-consistency is employed to assess the need for retrieval? When the model shows strong output consistency through self-consistency, retrieval may not be needed. Would this method be simpler and more effective than calculating self-aware uncertainty?

4. From Table 3, it seems that just using the LLM-generated rationale can achieve results close to those that require external retrieval. Does this imply that the knowledge retrieval in this method is not particularly effective, and the retrieved knowledge does not provide sufficient performance gains for the reasoning results?

5. Even if the LLM-generated rationale has low uncertainty, it may still represent a strong hallucination. How can this issue be avoided to better filter and re-rank the LLM-generated rationales?

[4] Li, Xingxuan, et al. "Chain-of-knowledge: Grounding large language models via dynamic knowledge adapting over heterogeneous sources." arXiv preprint arXiv:2305.13269 (2023).

---

> ### Author Response · Authors · 2024-11-24
> **Rebuttal [1/2]**
>
> Thanks for your detailed comments on our work! We are happy to address your concerns:
>
> ## Weakness 1
>
> > Some statements regarding the contributions in this paper are flawed. For example, the paper claims that existing adaptive RAG methods determine whether retrieval is needed solely based on the model’s output, which is clearly incorrect. Papers like Self-RAG[1] use the model’s in-context learning ability to judge whether additional knowledge retrieval is necessary, and during this process, the model naturally considers input prompts, including questions and retrieved knowledge. Additionally, many adaptive RAG methods re-rank retrieved knowledge during the integration process. For example, Self-RAG labels knowledge as "is_supportive," "is_related," and "is_useful." Similar work includes RankRAG[2], RagVL[3]. Therefore, the paper's claim that existing methods "neglecting to re-rank multiple retrieved knowledge and optimize the reasoning paths" is biased.
>
> Thanks for mentioning this! As you mentioned, Self-RAG adopts a similar strategy to classify retrieved documents. However, it does not perform detailed reranking operations. For RankRAG and RAGVL, as they are released within 4 month before the paper submission deadline, we are not aware of these works when we write this paper. They are considered as concurrent works according to the [policy](https://iclr.cc/Conferences/2025/FAQ) of ICLR 2025.
>
> We much appreciate you direct us to these works and will be more careful with our wording. In particular, we will add these works into the Related Works section and rewrite “these attempts” into “most of existing attempts” to make the paper more accurate.
>
>
> ## Weakness 2
>
> > The method proposed in this paper mainly consists of three processes: retrieval, re-ranking, and reasoning. The overall pipeline design is not novel, and the innovation of this paper lies primarily in using the model's self-aware uncertainty to evaluate the output of each process to decide when to retrieve, how to re-rank the retrieved knowledge, and whether to use the model's generated rationale or the retrieved information during reasoning. However, based on the experimental results in Table 3, using the proposed Self-aware Uncertainty Estimator does not provide a particularly significant improvement compared to using LN-Entropy, perplexity, or prompt to determine the LLM's uncertainty. Therefore, the innovation of this paper is limited, and the improvements in experiments may be more attributable to the effectiveness of the pipeline itself, while the innovation in the pipeline design is trivial.
>
> We are sorry for the confusion on the novelty of our paper and would like to explicitly state that, our novelty resides in:
>
> 1. Provide insights that reducing the uncertainty of LLMs could be a potential way to optimize RAG systems.
> 2. Provide a demonstration attempt to reducing the uncertainty of LLMs in the RAG systems via extracting uncertainty from the internal states, and show that this could be performed in 3 distinct stages (i.e., retrieval, re-ranking, and reasoning.) in RAG process.
>
> To prevent further misunderstanding, we are happy to further highlight our novelty in the paper.
>
>
> ## Weakness 3
>
> > Computing self-aware uncertainty seems to be a very time-consuming and computationally expensive process, yet the paper does not discuss the time and computational cost of the entire reasoning process. On the other hand, although this method is tuning-free, it can only be applied to local models. For closed-source LLMs, since their hidden layers are inaccessible, this method is not applicable, which weakens the advantage of this method in being tuning-free and affects its practical value.
>
>
> This is a good point! SeaKR benefits from the scaled-up computations during inference time, which is consistent with the observation from several recently released reasoning models, such as Open-o1.
>
> Our lack of computational resources drives us to optimize the whole system of SeaKR, and integrate vllm into the inference process. After our optimization, we find that SeaKR runs much faster than baselines. With 4 GeForce 3090, the average time to process a single question are shown below (in seconds). We plan to integrate this information into our paper.
>
> | Model       | 2WikiMultiHopQA  |  HotpotQA |
> |---|---|---|
> |FLARE     |  8.44         | 13.25 |
> |DRAGIN   | 29.75       | 19.25|
> |SeaKR      | 4.94           | 7.3|

---

> ### Author Response · Authors · 2024-11-24
> **Rebuttal [2/2]**
>
> ## Weakness 4
>
> > The experiment uses the F1 measure as an evaluation method, but in practice, when calculating F1, the model's output is matched word-by-word against the ground truth in an N-to-N fashion. This is not the standard F1 score evaluation method and does not reflect the method's retrieval performance. Moreover, this aspect was not clarified in the experimental design.
>
> The implementation of F1-measure in our experiments is consistent with baseline methods. You can check [**here**](https://github.com/oneal2000/DRAGIN/blob/main/src/data.py#L52) for the official evaluation codes of DRAGIN and [**here**](https://github.com/jzbjyb/FLARE/blob/main/src/datasets.py#L73) for FLARE. You may find that we all share exactly the same evaluation codes (direct copy-paste) for f1-score.
>
> Moreover, as we rerun the baselines in Table 1 and Table 2, we carefully check these setups to keep them consistent for fair comparison. We would like to further clarify these subtitles after revise our paper.
>
>
> ## Question 1
>
> > How long is the inference time required for the entire pipeline? Compared to the Prompt method, how much additional computation is needed when using the self-aware uncertainty estimator proposed in this paper? Is it applicable to larger LLMs?
>
> Thanks for your question! You may refer to our response to Weakness 3 for inference time of SeaKR.
>
>
> ## Question 2
>
> > Since the proposed self-aware uncertainty estimator can only be applied to local models, could the LLM be fine-tuned with an additonal adapter to better calculate uncertainty?
>
> Thanks for your question and the answer is yes! This is exactly our next work. We are trying to train the LLM to learn to reduce its uncertainty to enhance RAG performance. In particular, the uncertainty serves as a negative rewards in reinforment learning. However, we believe this is beyond the scope of this paper.
>
> ## Question 3
>
> > When determining whether retrieval is necessary, could a simpler and more effective approach be used, like Chain of Knowledge[4], where self-consistency is employed to assess the need for retrieval? When the model shows strong output consistency through self-consistency, retrieval may not be needed. Would this method be simpler and more effective than calculating self-aware uncertainty?
>
> Thanks for mentioning this. It has been shown that self-consistency within the internal states provides better approximation for model’s uncertainty than self-consistency in the output space [1-3].
>
>
> ## Question 4
>
> > From Table 3, it seems that just using the LLM-generated rationale can achieve results close to those that require external retrieval. Does this imply that the knowledge retrieval in this method is not particularly effective, and the retrieved knowledge does not provide sufficient performance gains for the reasoning results?
>
> We are sorry for the confusion. “Rationales-only” in Table 3 refers to the rationales generated by LLMs when they digest the retrieved documents. The are opposed to “Documents-only”, which means that the retrieved documents are not digest during the retrieval process and directly passed to the LLM for reasoning.
>
> In other worlds, the key information in the retrieved documents are distilled into the rationles.
>
>
> ## Question 5
>
> > Even if the LLM-generated rationale has low uncertainty, it may still represent a strong hallucination. How can this issue be avoided to better filter and re-rank the LLM-generated rationales?
>
> Thanks for the question! Evidence from diverse parties shows that pre-trained LLMs are mostly well-calibrated, which means that uncertainty could serve as a good estimator for model generated errors.We admit that it is still difficult to prune all hallucinations. To this end, we may need more reliable knowledge sources and more powerful LLMs. However, this is beyond the scope of this paper.
>
>
> [1] Chen C, Liu K, Chen Z, et al. INSIDE: LLMs' Internal States Retain the Power of Hallucination Detection[C]//ICLR 2024.
>
> [2] Wang X, Wei J, Schuurmans D, et al. Self-Consistency Improves Chain of Thought Reasoning in Language Models[C]//ICLR 2023
>
> [3] Zhang X, Yao Z, Zhang J, et al. Transferable and Efficient Non-Factual Content Detection via Probe Training with Offline Consistency Checking[C]. ACL 2024.
>
> [4] Anthropic. Language Models (Mostly) Know What They Know.
>
> [5] OpenAI. Language Models are Few-Shot Learners.

---

### Official Review · Reviewer_NzWJ · 2024-11-03

**Soundness:** 2
**Presentation:** 2
**Contribution:** 2
**Rating:** 3
**Confidence:** 4

**Summary:**

This paper proposes an adaptive RAG model that uses a multi-step reasoning strategy and dynamically decides whether to retrieve external knowledge based on an uncertainty estimator. The experiments show significant improvement over selected baselines. However, there are some flaws in paper writing, model design, and experiments. More details can be found in the weaknesses.

**Strengths:**

1. Using hidden states of the last token and the corresponding Gram matrix by sampling multiple generations to measure the generation uncertainty is novel compared to previous probability-based uncertainty measurements.
2. Experiments show significant improvement over selected baselines.

**Weaknesses:**

1. The organization and writing of the methodology introduction are a little bit confusing.
The first paragraph of Section 3 notes that the self-aware uncertainty estimator is the most important module. However, this paragraph only uses a symbol $U(c)$ to represent it without some clear introduction about its input/output and target. Then, in Sections 3.1, 3.2, and 3.3, the paper directly uses this symbol and only detailedly introduces $U(c)$ in Section 3,4. This writing style may lead to some confusion when reading the paper. I recommend first introducing your most important module, especially when it will be frequently utilized in other modules.

2. Lack of demonstrations of model design details.
In Section 3.1, what is the difference between "the pseudo-generated rationale" and "generated rationale"? How a query is generated specifically? More function, detailed introduction, and the corresponding motivation of model designs need to be provided to clarify your method process.

3. The calculation details of the "Gram matrix" can be provided to enrich the technique details of the paper.

4. The calculation of the "Gram matrix" relies on the multi-generation of LLMs. For each generation, the "<EOS>"'s hidden state still relies on the previously generated tokens. Does this process still be influenced by token sampling probabilities?

5. Since most of the modules are prompt-based, the stability of the proposed method is difficult to guarantee.

6. The selected search engine, BM25, is too old. A more advanced dense retriever or hybrid retriever could be utilized.

**Questions:**

1. Why the Section 3.1 is called "Self-aware retrieval"? It doesn't demonstrates more about retrieval process, but focuses on the generation of the next rationale and the next query.
2. For the "Query Generation" in Section 3.1, what does the symbol "r" refer to? Should it be "r_s"?
3. Why "Gram matrix" can reliably measure the uncertainty of LLM on its generation? Are there some related analyses or theoretical proven?
4. Why evaluations are conducted on sampled test instances? It leads to different model performance values in Tables 1, 2, and 3, which is very confusing.

---

> ### Author Response · Authors · 2024-11-24
> **[1/2] Rebuttal**
>
> Thanks for your constructive comments on our work! We would like to address your concerns, comments and questions below:
>
> ## Weakness 1
>
> > The organization and writing of the methodology introduction are a little bit confusing. The first paragraph of Section 3 notes that the self-aware uncertainty estimator is the most important module. However, this paragraph only uses a symbol U(c)  to represent it without some clear introduction about its input/output and target. Then, in Sections 3.1, 3.2, and 3.3, the paper directly uses this symbol and only detailedly introduces U(c)  in Section 3,4. This writing style may lead to some confusion when reading the paper. I recommend first introducing your most important module, especially when it will be frequently utilized in other modules.
>
>
> Thanks for your suggestion! We indeed have a discussion among all the authors on where to place the details of implementing U(c). The reason why we introduce U(c) until Section 3.4 is because we believe that it is more important to have the authors know the functionality of U(c) first and not to obstruct understanding of the adaptive RAG process with too much detail.
>
> We greatly appreciate your suggestions from the angle of a reader and decide to move the introduction of U(c) to Section 3.1.
>
> ## Weakness 2
>
> > Lack of demonstrations of model design details. In Section 3.1, what is the difference between "the pseudo-generated rationale" and "generated rationale"? How a query is generated specifically? More function, detailed introduction, and the corresponding motivation of model designs need to be provided to clarify your method process.
>
> Thanks for suggesting further to clarify the model design details.
>
> In particular, “the pseudo-generated rationale” refers to a true model generation. However, the generated rationale is not directly used in the reasoning process. It is only used to help form queries. Thus, we term it as “pseudo-generated” rationales.
>
> I would like to further clarify our rationales on the query generation process. We ask the LLM to try to answer the question by generating one rationale, and identify uncertain tokens in that generation. In this way, the query, which is used as the input to the search engine, is generated by removing tokens with high uncertainty from the pseudo-generated rationales. We hope that information from the search engine would fill the removed uncertain tokens, as we stated in Line 187.
>
> I hope this explanation answers your concern and would like to further enrich our paper by incorporating the above explanations in Section 3.1.
>
> ## Weakness 3
>
> > The calculation details of the "Gram matrix" can be provided to enrich the technique details of the paper.
>
> Thanks for suggesting to elaborate on the details of the Gram matrix. For a set of representations, which we denote as $\{\mathbf{H}[i]\}$, the $i, j^{\text{th}}$ element in the Gram matrix $G$ is calculated as the inner product between $\mathbf{H}[i]$ and $\mathbf{H}[j]$.
>
> Formally, we have $G_{i,j}=\langle \mathbf{H}[i], \mathbf{H}[j] \rangle$. We would like to further explain the calculation details of the Gram matrix after revision.
>
>
> ## Weakness 4
>
> > The calculation of the "Gram matrix" relies on the multi-generation of LLMs. For each generation, the "<EOS>"'s hidden state still relies on the previously generated tokens. Does this process still be influenced by token sampling probabilities?
>
>
> That is a good question. Our process to calculate the Gram determinant is to estimate the expectation of the Gram determinant via Monte-Carlo sampling. Thus, it is influenced by the sampling process and the variance is influenced by sampling times $k$.
>
> We empirically use $k=20$ and find that further enlarging this value brings marginal benefits. You can check Figure 3 for more details. Thus, when $k \ge 20$, we can say that the Gram matrix is almost not influenced by the token sampling process.
>
>
> ## Weakness 5
>
> > Since most of the modules are prompt-based, the stability of the proposed method is difficult to guarantee.
>
>
> Thanks for mentioning this!. We submitted the prompts that are used to produce the results in the supplemented material to facilitate reproducibility. We will open-source our prompts if this paper is published. For more details, you may check Appendix D.
>
>
> ## Weakness 6
>
> > The selected search engine, BM25, is too old. A more advanced dense retriever or hybrid retriever could be utilized.
>
> Thanks for mentioning that we can further optimize the search engine. We agree that BM25 is not the state-of-the-art search engine. We choose BM25 as our search engine because this is **in line with baseline adaptive RAG methods**. We strictly control variables in the experiment setup. We would like to further highlight this after revision.

---

> ### Author Response · Authors · 2024-11-24
> **[2/2] Cont.**
>
> ## Question 1
>
> > Why the Section 3.1 is called "Self-aware retrieval"? It doesn't demonstrates more about retrieval process, but focuses on the generation of the next rationale and the next query.
>
>
> We call the module introduced in Section 3.1 *Self-aware Retreival* because this is the only module that interacts with the search engine directly.
>
> ## Question 2
>
> > For the "Query Generation" in Section 3.1, what does the symbol "r" refer to? Should it be "r_s"?
>
>
> Thanks for mentioning this! We will fix the typo after revision.
>
> ## Question 3
>
> > Why "Gram matrix" can reliably measure the uncertainty of LLM on its generation? Are there some related analyses or theoretical proven?
>
>
> We follow the theory of INSIDE [1] to use the Gram determinant to measure the uncertainty of LLM because it can be used to measure the inconsistency among multiple generations in the space of the hidden states. It has been shown that the uncertainty of LLMs can be reflected by models’ inconsisntencies [2][3].
>
> ## Question 4
>
> > Why evaluations are conducted on sampled test instances? It leads to different model performance values in Tables 1, 2, and 3, which is very confusing.
>
> Thanks for the question! We conduct evaluations on sampled test instances because of limited computational resources. In particular, the main results in Table 1 and Table 2, which involve the comparison with other baselines, are not evaluated on sampled test instances. For Table 3, which does not involve the comparison with other baselines, we follow the experiment setup of FLARE (also one of our baselines) to sub-sample 500 questions, which greatly alleviate our burden to apply for more computational resources.
>
>
>
>
>
> [1] Chen C, Liu K, Chen Z, et al. INSIDE: LLMs' Internal States Retain the Power of Hallucination Detection[C]//ICLR 2024.
>
> [2] Wang X, Wei J, Schuurmans D, et al. Self-Consistency Improves Chain of Thought Reasoning in Language Models[C]//ICLR 2023
>
> [3] Zhang X, Yao Z, Zhang J, et al. Transferable and Efficient Non-Factual Content Detection via Probe Training with Offline Consistency Checking[C]. ACL 2024.

---

### Official Review · Reviewer_8hhY · 2024-11-05

**Soundness:** 3
**Presentation:** 3
**Contribution:** 3
**Rating:** 8
**Confidence:** 4

**Summary:**

The paper introduces SEAKR, a novel approach for adaptive Retrieval-Augmented Generation (RAG) that utilizes self-awareness within large language models (LLMs) to optimize when and how to retrieve external knowledge. The primary objective of SEAKR is to mitigate hallucination in LLMs by dynamically determining the necessity of external retrieval based on self-aware uncertainty detected within the model’s internal states. SEAKR integrates retrieved information using self-aware re-ranking, which selects knowledge snippets that effectively reduce uncertainty, and self-aware reasoning, which iteratively refines the retrieval strategy to address complex queries. Experimental results on both complex and simple question-answering tasks demonstrate that SEAKR significantly outperforms existing adaptive RAG methods by adapting retrieval and integration strategies to the model’s self-identified knowledge gaps.

**Strengths:**

- The writing is easy to follow and the logic is clear.
- The authors provide extensive experimental results across both complex and simple question-answering tasks , validating the robustness and superiority of SEAKR over state-of-the-art approaches.
- SEAKR is one of the first methods to use self-aware uncertainty within the internal states of LLMs to guide retrieval decisions. By utilizing the model’s internal consistency, SEAKR ensures that retrieval is only triggered when necessary, addressing the inefficiencies and potential conflicts caused by indiscriminate retrieval.

**Weaknesses:**

- Sensitivity to Uncertainty Thresholds: SEAKR’s reliance on a carefully chosen uncertainty threshold for retrieval introduces challenges in generalizability. Determining the optimal threshold can be difficult, and miscalibration may lead to over-retrieval or under-retrieval. This sensitivity could be problematic if SEAKR were applied across different domains with varying uncertainty levels in LLMs, potentially requiring frequent recalibration.
- Challenges in Numerical and Logical Reasoning: The paper notes that SEAKR’s improvement on datasets requiring numerical reasoning (e.g., IIRC) is limited. This suggests that while SEAKR can handle general knowledge queries well, it may not perform as effectively on questions that involve mathematical calculations, precise date comparisons, or logical deductions. Future iterations might need specialized handling of these reasoning types.

**Questions:**

- Why self-aware uncertainty from the internal state of LLMs can more accurately determines the knowledge demand compared with the output?
- What strategies could be introduced to better handle tasks that require numerical or logical reasoning?

---

> ### Author Response · Authors · 2024-11-24
>
> We much appreciate your positive attitude towards SeaKR and your valuable feedback! We are happy to address your concerns and questions:
>
> ## Weakness 1
> > Sensitivity to Uncertainty Thresholds: SEAKR’s reliance on a carefully chosen uncertainty threshold for retrieval introduces challenges in generalizability. Determining the optimal threshold can be difficult, and miscalibration may lead to over-retrieval or under-retrieval. This sensitivity could be problematic if SEAKR were applied across different domains with varying uncertainty levels in LLMs, potentially requiring frequent recalibration.
>
>
> Thanks for pointing out that SeaKR is sensitive to the threshold of uncertainty. We address this issue by utilizing a dev set and treat the threshold as a hyperparameter. The optimal threshold can be found via choosing the best-performing threshold on the dev set, as we explain at Line 297.
>
> When we conduct our experiments, we share the same threshold across all 6 datasets. The threshold $\delta$ is searched on the same dev-set. We acknowledge that further tuning $\delta$ on domain-specific dataset will bring extra benefit.
>
>
> ## Weakness 2 & Question 2
>
> > Challenges in Numerical and Logical Reasoning: The paper notes that SEAKR’s improvement on datasets requiring numerical reasoning (e.g., IIRC) is limited. This suggests that while SEAKR can handle general knowledge queries well, it may not perform as effectively on questions that involve mathematical calculations, precise date comparisons, or logical deductions. Future iterations might need specialized handling of these reasoning types.
>
> > What strategies could be introduced to better handle tasks that require numerical or logical reasoning?
>
> Thanks for further highlighting our observations on Numerical and Logical Reasoning tasks with SeaKR!
>
> Addressing symbolic reasoning tasks is indeed within our future plan on SeaKR. We are happy to share our plan here. In particular, we are exploring two research questions with SeaKR: (1) Internalize the uncertainty estimation capability via model training and (2) boosting model’s capability on symbolic reasoning tasks.
>
> The two research questions can be both addressed by reinforcement learning, which is widely hypothesized to be the technique behind Open-o1. We place uncertainty estimation in the action space and encourage the LLM to perform self-reflection to better answer numerical and logical reasoning tasks. However, we believe this is beyond the scope of this paper.
>
>
>
> ## Question 1
>
> > Why self-aware uncertainty from the internal state of LLMs can more accurately determines the knowledge demand compared with the output?
>
> Thanks for raising the question. One of the advantages of SeaKR is that it estimates the uncertainty from models’ internal states, rather than their outputs. It has been shown by previous works that the internal states are more faithful to the true belief of LLMs, while models may fail to honestly judge their internal states. We plan add more discussion on this after revision to further highlight the rationale of our method.

---

> > ### Comment · Reviewer_8hhY · 2024-11-28
> >
> > Thank you for your detailed response. I have reviewed your rebuttal and found that it sufficiently addresses my concerns. As a result, I have updated my score accordingly.

---

> > > ### Author Response · Authors · 2024-12-03
> > > **Thanks for your comment**
> > >
> > > Thank you for your active feedback and engagement during the rebuttal process! We much appreciate your endorsement and would be happy to continually evolve our work.

---

### Official Review · Reviewer_FqGE · 2024-11-05

**Soundness:** 3
**Presentation:** 2
**Contribution:** 3
**Rating:** 6
**Confidence:** 4

**Summary:**

This paper introduces a self-aware knowledge retrieval approach for RAG. Instead of retrieving information for all queries, this technique triggers retrieval only when the language model is uncertain about its answer. The uncertainty is measured by the correlation (based on the Gram matrix) of the last token representations from multiple parallel samplings. The paper also proposes using the uncertainty score to rerank the retrieved documents and combines iterative COT reasoning with retrieval. Experiments on multiple datasets show that the proposed method consistently outperforms previous RAG methods.

**Strengths:**

- The proposed method is well-motivated and performs effectively on benchmarks.

- The new uncertainty measurement is interesting and empirically outperforms previous methods like Perplexity or energy-based approaches.

- This paper conducts extensive experiments, and the results are insightful and support its main claims.

**Weaknesses:**

- The proposed method includes using the consistency of parallel sampling (with a sample time of 20) to determine the uncertainty of the LLM. This is performed at each retrieval step, which may increase computational costs during inference. It would be beneficial to compare the inference latency of these methods or evaluate them under a similar inference budget.

- I suggest moving Section 3.4 to an earlier position in the paper, as it appears to be the key proposal and is referenced in Sections 3.1, 3.2, and 3.3. Additionally, the current Section 3.4 is somewhat vague and could benefit from more detail.

- The paper finds that more retrieval steps decreases model performance, as shown in Table 3. This may be due to the use of a weak retriever (BM25) and the recall of only a few documents (top-3). The paper should also compare against a stronger retriever and with a larger number of recalled documents, as is common in most RAG settings.

**Questions:**

None

---

> ### Author Response · Authors · 2024-11-24
>
> Thanks for your positive attitude towards our work and your constructive feedback! We would like to address your concerns towards SeaKR below:
>
> ## Weakness 1
> > The proposed method includes using the consistency of parallel sampling (with a sample time of 20) to determine the uncertainty of the LLM. This is performed at each retrieval step, which may increase computational costs during inference. It would be beneficial to compare the inference latency of these methods or evaluate them under a similar inference budget.
> Indeed, SeaKR involves increased computational costs to benefit from inference time scaling-up, which is in line with several recent discoveries[1,2]. We are aware of that and address this issue by integrating SeaKR with vllm---a highly efficient inference engine for LLMs---to reduce inference latency. We mentioned this at Line 296.
> As you suggested, we compare the latency between SeaKR and previous adaptive RAG methods (FLARE, DRAGIN)  for Complex QA tasks, which potentially involves more tokens to generate than Simple QA tasks. The average times to process a single question in seconds with 4$\times$ Geforce 3090 are shown below. We would like to integrate this information after revision.
>
> | Model       | 2WikiMultiHopQA  |  HotpotQA |
> |---|---|---|
> |FLARE     |  8.44         | 13.25 |
> |DRAGIN   | 29.75       | 19.25|
> |SeaKR      | 4.94           | 7.3 |
>
>
> ## Weakness 2
> > I suggest moving Section 3.4 to an earlier position in the paper, as it appears to be the key proposal and is referenced in Sections 3.1, 3.2, and 3.3. Additionally, the current Section 3.4 is somewhat vague and could benefit from more detail.
>
> Thanks for your suggestion! We indeed considered introducing the uncertainty estimator first. We finally decided to place the details of the uncertainty estimator in Section 3.4 because we want to make the audiences focus on the overall framework of SeaKR. In this case, the reader only needs to understand the functionality and input/output format of the uncertainty estimator.
> We greatly appreciate your suggestion from the perspective of readers. We introduce  the uncertainty estimator after revising the paper.
>
>
> ## Weakness 3
> > The paper finds that more retrieval steps decreases model performance, as shown in Table 3. This may be due to the use of a weak retriever (BM25) and the recall of only a few documents (top-3). The paper should also compare against a stronger retriever and with a larger number of recalled documents, as is common in most RAG settings.
>
>
> Thanks for suggesting to utilize a more powerful search engine. We choose to implement SeaKR with BM25 because this is consistent with baseline adaptive RAG methods, i.e., FLARE and DRAGIN. These variables are strictly controlled.
>
> As SeaKR does not optimize the search engine, it is orthogonal to any other search engines and we believe that it could directly benefit from more capable retrievers, same as baselines.
>
>
>
> [1] Snell C, Lee J, Xu K, et al. Scaling LLM test-time compute optimally can be more effective than scaling model parameters[J]. arXiv preprint arXiv:2408.03314, 2024
>
> [2] Brown B, Juravsky J, Ehrlich R, et al. Large language monkeys: Scaling inference compute with repeated sampling[J]. arXiv preprint arXiv:2407.21787, 2024

---

> ### Comment · Reviewer_FqGE · 2024-12-01
>
> Thank you for your response and the new results!
>
> I think the referenced research on inference time scaling-up highlights the importance of comparing methods under the same compute budget. It's good to see more analysis in this regard.
>
> About the paper structure, I am confused when I read about an important function that was not clearly introduced.
>
> As for retrieval, I understand the comparison is fair, but I am concerned that a stronger retriever might yield different results. BM25 relies only on lexical overlap and could behave differently compared to dense retrievers. Therefore I am interested in the results of dense retriever.

---

> > ### Author Response · Authors · 2024-12-03
> > **Thanks for your comment!**
> >
> > We much appreciate your engagement during the rebuttal period and constructive suggestions! We would be happy to test SeaKR with a dense retriever, like DPR if time permits.

---

### Author Response · Authors · 2024-11-24
**General Response**

We much appreciate all the reviewers for their hard work and invaluable advise to further polish our paper!

We are happy to address all your concerns and questions point by point and plan to revise our paper accordingly within the coming 1 or 2 days.

---

### Meta-Review · Area_Chair_5QXD · 2024-12-21

**Metareview:**

This paper proposes a novel RAG method, which retrieves knowledge only when the LLM is uncertain about the results. The authors conduct many experiments to demonstrate that the proposed method consistently outperforms previous RAG methods.

In general, the reviewers think that:

(1) The idea of the paper is interesting.
(2) The experiment is comprehensive and can demonstrate that the proposed model can outperform previous methods.

However, the reviewers also  present many concerns:

(1) The proposed model can increase the computational cost.
(2) Lack of demonstrations of model design details.
(3) The model is mostly based on prompts, which may lead to instability.
(4) Some experiments are not convincing (the old search engine BM25).

**Additional Comments On Reviewer Discussion:**

In the rebuttal period, the authors can address several concerns raised by the reviewers. However,  reviewer 1gAL still has doubts about the novelty of the paper. In addition, the reviewer of the highest score (8) does not present opinions to strongly suggest accepting this paper.  Thus, I tend to reject the paper.

---

### Decision · Program_Chairs · 2025-01-22

Reject